# The Interaction of Two Widely Used Endodontic Irrigants, Chlorhexidine and Sodium Hypochlorite, and Its Impact on the Disinfection Protocol during Root Canal Treatment

**DOI:** 10.3390/antibiotics12030589

**Published:** 2023-03-16

**Authors:** Dirk-Joachim Drews, Anh Duc Nguyen, Antje Diederich, Christian Ralf Gernhardt

**Affiliations:** 1Private Dental Practice, 69469 Weinheim, Germany; 2University Outpatient Clinic for Conservative Dentistry and Periodontology, Martin-Luther-University Halle-Wittenberg, 06112 Halle, Germany

**Keywords:** chlorhexidine digluconate, para-chloroaniline, precipitation, root canal irrigants, sodium hypochlorite

## Abstract

In recent years, sodium hypochlorite and chlorhexidine digluconate have been the gold standard of irrigation solutions utilized within the disinfection protocol during root canal treatments. Nowadays, it is known that, during chemical disinfection of the root canal, consecutive application of sodium hypochlorite and chlorhexidine digluconate leads to the formation of an orange-brown precipitate. This precipitate is described as being chemically similar to para-chloroaniline, which is suspected to have cytotoxic and carcinogenic effects. Concerns also exist regarding its influence on the leakage of root canal fillings, coronal restorations, and tooth discoloration. The purpose of this article is to review the literature on the interaction of sodium hypochlorite and chlorhexidine digluconate on the tooth and its surrounding tissues, and to discuss the effect of the precipitate formed during root canal treatment. We further address options to avoid the formation of the precipitate and describe alternative irrigation solutions that should not interact with sodium hypochlorite or chlorhexidine digluconate.

## 1. Introduction

Complete cleaning and disinfection of the root canal system are considered mandatory for long-term success in root canal treatment [1,2]. However, even after thorough mechanical cleaning, residual pulp tissue, bacteria, and dentin debris can remain in the root canal system [3,4]. Therefore, a variety of irrigating solutions are used in combination with the mechanical processing, such as sodium hypochlorite (NaOCl), chlorhexidine digluconate (CHX) [5], 17% ethylenediaminetetraacetic acid (EDTA), citric acid (CA), BioPure^®^ MTAD^®^ (Dentsply Tulsa Dental Specialties, Tulsa, OK, USA), and 37% phosphoric acid (PA) [6], as well as etidronate, alexidine (ALX), and Octenisept^®^ (Schülke & Mayr, Norderstedt, Germany) [7]. Following internationally accepted quality guidelines, the main goals of irrigation are: eliminating microorganisms, flushing out debris, lubricating root canal instruments, and dissolving organic debris. Therefore, the used irrigation solution should preferably have disinfectant and organic-debris-dissolving properties, whilst not irritating the periradicular tissues [8]. For this purpose, sodium hypochlorite and chlorhexidine digluconate are widely recommended and well accepted in endodontics [9,10].

Unfortunately, endodontic irrigation solutions may interact chemically with each other during an alternating irrigation technique, potentially forming unwanted by-products, which may be toxic or cause allergic reactions [7]. Sodium hypochlorite and chlorhexidine are the best known and, at least in recent years, most frequently recommended irrigating solutions used for eliminating residual bacteria in chemo-mechanical root canal processing [5,6]. The undesirable adverse effects, after sodium hypochlorite and chlorhexidine interaction, of building precipitates are known, published and discussed controversially [11]. However, it is recommended that, until this precipitate is studied further, its formation should be avoided by removing the NaOCl before placing CHX into the canal [11]. Since 2006, the number of articles in PubMed concerning the interaction of NaOCl and CHX have grown significantly, and the topic was greatly debated [12,13,14,15,16]. Therefore, the aim of the present review is to summarize and discuss recently published papers focusing on the different outcomes regarding the interactions between sodium hypochlorite and chlorhexidine. Furthermore, based on the results of the review, the possible impact for the clinical disinfection protocol in endodontic therapy is summarized.

### 1.1. Sodium Hypochlorite (NaOCl)

Sodium hypochlorite (Figure 1) is the most used irrigating solution in endodontics, because its mechanism of action causes biosynthetic alterations in cellular metabolism and phospholipid destruction, the formation of chloramines that interfere in cellular metabolism, oxidative action with irreversible enzymatic inactivation of bacteria, and lipid and fatty acid degradation [17].

Sodium hypochlorite (NaOCl) is the most common irrigant used in root canal treatments. NaOCl is an effective tissue solvent and antimicrobial agent. It is usually used in a concentration range from 0.5 to 8.25% [18,19,20]. Its germicidal ability is related to the formation of hypochlorous acid when in contact with organic debris. In high concentrations, NaOCl is toxic and can cause inflammation in the periapical tissues [21], whereas in low concentrations, it is ineffective against specific microorganisms. NaOCl is not a substantive antimicrobial agent; it tends to discolor and corrode surgical instruments; and it has a very unpleasant odor [11].

### 1.2. Chlorhexidine (CHX)

Chlorhexidine digluconate (CHX) is the gluconate salt form of chlorhexidine, a biguanide compound used as an antiseptic agent with topical antibacterial activity (Figure 2). Chlorhexidine digluconate is positively charged and reacts with the negatively charged microbial cell surface, thereby destroying the integrity of the cell membrane. Subsequently, chlorhexidine gluconate penetrates into the cell and causes a leakage of intracellular components, leading to cell death. Since gram-positive bacteria are more negatively charged, they are more sensitive to this agent [22].

Chlorhexidine digluconate (CHX) can be used as a complement to increase the antibacterial action of NaOCl solutions during root canal preparation. CHX shows similar antimicrobial effects to sodium hypochlorite [23,24] in vitro and possesses a lower toxicity [25,26]. A disadvantage compared to NaOCl is its lack of ability to dissolve vital and necrotic tissue [27].

Chlorhexidine digluconate is a broad-spectrum antibacterial agent with substantivity to tooth structures, i.e., it binds to the hydroxyapatite of the enamel and dentin or to anionic groups of glycoproteins, is slowly released and, due to the moderate concentration decrease, its antibacterial effects are prolonged for an extended period of time [28].

### 1.3. Proteolysis

When NaOCl and CHX are mixed, NaOCl dissociates into different ions (H^+^, O^2−^, and Cl^−^). The chloride group then reacts with the chlorhexidine molecule in the guanine group (NH). This leads to the formation of chlorhexidine chloride (N^+^ and Cl^−^). In this reaction, the formation of an orange-brown precipitate is described. This precipitate contaminates the dentin and adheres to the canal walls [29]. Furthermore, CHX is a dicationic acid and has the ability to donate protons, whereas NaOCl is alkaline and can absorb protons from the dicationic acid. This proton exchange leads to the formation of a neutral and insoluble precipitate [11,30,31]. A color change due to the reaction can already be seen from a concentration of 0.023% NaOCl and the formation of the precipitate from 0.19% NaOCl by means of X-ray Photoelectron Spectroscopy (XPS), and the absolute amount by means of Time-of-Flight Secondary Ion Mass Spectrometry (ToF-SIMS) [11].

While the undesirable effects of the initial developing substances have been well studied and are classified as acceptable [5], the precipitate with regard to its ingredients and undesirable effects still gives rise to discussions [32,33].

Figure 3 demonstrates microtubes filled with 2% CHX mixed with different concentrations of NaOCl. The first microtube is a control sample with 2% CHX alone. From left to right, a color change, which becomes brighter as the concentration of NaOCl decreases, can be observed.

## 2. Materials and Methods

An unlimited search in all fields of the PubMed database (https://pubmed.ncbi.nlm.nih.gov/, accessed on 4 October 2022) was carried out through the website of the National Center for Biotechnology Information (NCBI), utilizing the combination of the Medical Subject Headings (MeSH terms) “sodium hypochlorite” (NaOCl) AND “chlorhexidine” (CHX) and yielded 955 results from the years 1974–2022. Specifying the search term to “chlorhexidine AND sodium hypochlorite AND interaction”, 64 publications remained from the original result. By individually reviewing the references and abstracts of these 64 publications the keywords “precipitate” and “para-chloroaniline” were regularly found in the keywords of the relevant articles (Figure 4).

Finally, the search term was further expanded to include “chlorhexidine AND sodium hypochlorite AND (interaction OR precipitate OR chloroaniline)”, which resulted in a selection of 88 articles that included the manually determined references. The abstracts of all articles of the final online search result were evaluated and 25 articles that showed no relevance to the question were sorted out (Table 1, Figure 4).

## 3. Results

The 63 papers included in this review are listed in Table 2, where the title and objective were summarized. These publications were read in full and evaluated.

58 publications (49 studies, 8 reviews, 1 short communication) were relevant to the topic; another 5 were excluded after reading the full texts. Sources to which the research publications referred were included if they were relevant to the topic, even if the date of their publication was before 1994.

## 4. Discussion

The PubMed search found 63 publications, 58 of which were relevant. After full text analysis, 49 were studies that have been published since 2006 in the medical and especially in the dental-endodontic field, which have dealt with the interaction of NaOCl and CHX. Eight reviews with different focuses giving an overview of the state of knowledge at the date of publication were also selected. Furthermore, one article made recommendations on how to avoid formation of the precipitate, and thus was included.

### 4.1. Methodology

The 1998 study by Kuruvilla and Kamath [83] indicated that the alternating use of NaOCl and CHX reduces the microbial flora to a greater percentage (84.6%) than the use of NaOCl (69.4%) or CHX (70%) on its own. In order to optimize the tissue-dissolving properties of NaOCl and the antiseptic properties of the CHX against gram-positive germs, it was considered to use a combination of both irrigation solutions. However, by mixing the two solutions, for example through consecutive use in the root canal, a peach-colored to brown precipitate is formed [7], which is difficult to remove [10,13,16,17]. It is undisputed in the literature that the precipitate forms due to the acid–base reaction of NaOCl and CHX. The exact composition and, in particular, the question of whether the precipitate contains para-chloroaniline (PCA), motivated studies in the period from 2007 to 2021. Controversial views on the suitability of test methods for the analysis of the precipitate [13,78,80] and partly contradicting test results from the same test methods [47,59,76] leave doubts as to whether free PCA arises from the reaction of NaOCl and CHX [32]. However, recent studies emphasize the use of multiple non-destructive test methods and always examine 98% PCA as a comparison group, and they could not detect any free PCA in the precipitate [47,55]. In a review carried out by two independent authors on the basis of 13 included articles from different databases, Khatib et al. [33] concluded that the brown precipitate, which forms after mixing NaOCl and CHX, may contain a proportion of para-chloramide rather than free PCA and that PCA may be the by-product of the breakdown of highly concentrated CHX. It is also disputed whether PCA has mutagenic potential. While Gomes et al. [6] were citing publications from 1986 and 1995 according to which PCA was found to be mutagenic in microorganisms, Patil et al. [54] found no significant difference in the mutagenicity of the precipitate and the comparison group.

### 4.2. Toxicity

Regarding the toxicity of the precipitate, Cintra et al. [63] found a short-term increased toxicity compared to the starting substances, while Vouzara et al. [53] identified a predominantly antagonistic effect in the combination of NaOCl and CHX, indicating that the precipitate was less toxic than the starting substances. Surrender et al. [51] found the precipitate to be less toxic than either NaOCl or CHX alone. Furthermore, Jeong, Sarmast, Terlier, van der Hoeven, Holland and Parikh [35] all concluded that the precipitate has a toxic effect against human gingival fibroblasts, but highly concentrated NaOCl has an even greater cytotoxic effect. Nocca et al. [14] also observed a lower mortality of fibroblast cells to which the precipitate was applied than in those treated with the supernatant.

Marchesan et al. [84] evaluated the metals present in the precipitate of NaOCl and CHX by means of atomic absorption spectrophotometry and identified statistically significant proportions of copper (Cu), tin (Zn), iron (Fe), manganese (Mn), magnesium (Mg) and calcium (Ca). Siddique et al. [16] found selenium (Se) with inductively coupled plasma mass spectrometry. A discoloration of enamel and dentin was found by Souza et al. [65] on bovine anterior teeth that were placed in CHX gel and NaOCl consecutively. Therefore, it could be concluded that the combined use of NaOCl and CHX solution can cause dentin discoloration during endodontic treatment.

### 4.3. Recommended Irrigation Protocol

In order to prevent the formation of precipitates when using NaOCl and CHX, an irrigation protocol that includes intermediate rinses has frequently been recommended. For example, Zehnder [5] recommended rinsing the root canals exclusively with NaOCl during the mechanical preparation, which he ascribed to “unique tissue-dissolving properties”. Before a final rinse with CHX recommended by him for chronic pulpitis and revisions, Zehnder [5] advised an intermediate rinse with EDTA or citric acid in order to prevent the formation of precipitates. It should be noted here that the root dentin can soften, if it is exposed to strong chelating agents, such as EDTA, for a long time [85]. Bueso et al. [34] used stereomicroscopic analysis to compare the effect of EDTA, distilled water and sodium thiosulfate (STS) as an intermediate rinse to prevent the formation of brown precipitates. In this context, 5% STS significantly reduced the intensity of brown precipitates, compared to no intermediate rinse. Alberto et al. [37] were able to demonstrate this effect ex vivo when CHX was added 10 min after the application of STS. Subsequent studies evaluated the endodontic irrigation regimen. The formation of precipitates was also demonstrated for the mixture of EDTA and NaOCl, but not for citric acid and CHX [72].

In addition, Mortenson et al. [13] found the least amounts of precipitate after intermediate flushing with 50% citric acid, compared to EDTA and saline. Intermediate rinsing with pure alcohol, distilled water, or saline solutions could also prevent or reduce the formation of precipitates [79]. However, a precipitate present in the root canal system represents a layer that occludes the dentinal tubules [12,30], is difficult to remove [29,79], and compromises the tightness of a root filling using AH 26 sealer (Dentsply Sirona, Konstanz, Germany) and gutta-percha proportionally to the amount of precipitate [61]. Whether the precipitate affects sealer adhesion has been controversially discussed: While Gupta et al. [70] came to the conclusion that the precipitate reduced the bonding capacity of an epoxy-based sealer (AH Plus^®^, Dentsply Sirona, Konstanz, Germany) significantly, it was subsequently shown that the adhesion of Resilon^®^-Epiphany SE obturation system (Pentron Clinical Technologies, Wallingford, CT, USA) was not affected by the precipitate. In addition, Magro et al. [58,62] found no correlation between the penetration depth of an epoxy sealer into the dentin and bond strength values between groups treated with or without CHX. However, the investigated CHX variants led to more precipitate in all root canal areas previously rinsed with NaOCl, though they did not reduce the bond strength of the sealer in the push-out test, which was traced back to the protocol for canal drying and covalent bonds between the sealer and the dentin surface [58,62].

Even by activating the rinsing solutions, the removal of the precipitate is only possible to a limited extent. However, it was found that activation of the chelating agents EDTA and citric acid, in particular using sonic (Eddy^®^, VDW, Munich, Germany) or ultrasound devices, is superior to syringe rinsing [32,52,57].

### 4.4. Alternative Irrigation Solutions

Furthermore, CHX alternatives were also considered and examined. The substitution of CHX by the herbal antimicrobial substances neem, tulsi, aloe vera, and garlic was not successful, as the amount of precipitate resulting from these substances in combination with NaOCl was a factor of 4–7.5 higher than that with CHX [42]. ALX, a substance from the biguanide family, similar to CHX, developed only a slightly yellowish color, but no precipitate formation with NaOCl [46]. In studies by Thomas et al. [41] the effectiveness of the combination of ALX and NaOCl against Enterococcus faecalis was not significantly higher than that of NaOCl alone. Nevertheless, the authors propagated that it can be used in 1% concentration as an alternative to CHX in the endodontic irrigation protocol if used for a sufficiently long time (>5 min). In contrast to the combination of NaOCl and CHX, Kim et al. [86] found no PCA in the mixture of NaOCl and ALX and considered it to be a CHX alternative because it is just as effective against all bacteria and fungi. Czopik [36] described a yellowish precipitate when mixing NaOCl and ALX, which could be identified as aliphatic amines by using the UHPLC-MS (ultra-high-performance liquid chromatography-mass spectrophotometry) method. Surender et al. [51] found a significantly higher effectiveness of NaOCl with ALX against Enterococcus faecalis than with the combination of NaOCl and CHX. Octenisept^®^, an octenidine-based preparation, which also contains 2% phenoxyethanol, led to a sparse, whitish deposits that partially closed the dentinal tubules and became transparent over time. Thaha et al. [50] saw potential for a combined application with NaOCl, but also a need for further investigations, for example, with regard to the effect on sealer adhesion to dentin. MTAD^®^, which contains 3% doxycycline, 4.25% CA, and 0.55% polysorbate, forms a green-yellow precipitate with NaOCl, the color of which changes to brown when exposed to light. Intermediate rinsing with ascorbic acid can prevent precipitation [7]. SmearOFF^TM^ (Vista Apex, Raxine, WI, USA) and QMix^®^ (Dentsply Sirona, Bensheim, Germany) are products that combine a biguanide and a chelator. After their application, the penetration depths of the sealer into the dentin were greater than those after sequential rinsing with 17% EDTA, saline solution, and CHX. While QMix^®^ is used after saline or distilled water (2-phase), SmearOFF^TM^ combines the intermediate rinse and the final rinse, which simplifies and shortens the rinsing protocol. According to the manufacturer, the use of SmearOFF^TM^ after NaOCl in the root canal does not lead to the formation of precipitates; this is also indicated by the sealer penetration depths. Since the manufacturers have not disclosed the formulation of the preparations, further studies on effectiveness and interactions are required [39].

The possibility of exchanging NaOCl in the combination of NaOCl and CHX was only considered possible by Buyukozer et al. [38]. Chlorine dioxide (ClO_2_) can be utilized as an alternative means of root canal irrigation instead of NaOCl, due to its antimicrobial activity, biocompatibility, and ability to dissolve organic tissue [87,88,89].

### 4.5. Clinical Impact on Endodontic Therapy

In 2023, the best possible cleaning and disinfection of the root canal system by means of chemomechanical preparation is still an indispensable prerequisite for the success of endodontic treatment [2,90,91].

The desirable properties of the various irrigation solutions are:-Dissolution of necrotic and vital tissue;-Effectiveness against bacteria;-Effectiveness against fungi;-Neutralization of endotoxins;-Opening of the dentinal tubules;-Removal of iatrogenic impurities;-Economic efficiency;-Practicality.

Unwanted properties are:-Irritation of neighboring tissues;-Cytotoxicity;-Mutagenicity;-Changes in the color of dentin or tooth enamel;-Occlusion of the dentinal tubules;-Undesirable interactions with other endodontic irrigating solutions and materials.

Since no irrigation solution is known that combines all the necessary properties and can be solely applied clinically, different solutions are used consecutively [92]. Certain combinations can interact with each other and result in undesirable effects or by-products. When in contact with each other in the root canal system, NaOCl and CHX interact in an acid–base reaction, forming an orange-brown precipitate, which has undesirable effects. The occlusion of the dentinal tubules [12,30] is indisputable and, depending on the sealer, can have a negative effect on its adhesive force or tightness [29,61,79]. The dyes of the precipitate can discolor the tooth substances, particularly the dentin.

Although it can be considered unlikely that the precipitate of NaOCl and CHX contains free para-chloroaniline, a substance that is suspected of being mutagenic, it is important to avoid precipitation in the root canal. After formation, the complete removal of the precipitate from the root canal system is difficult or impossible, even with advanced methods of activating irrigation solutions with sound, ultrasound, or laser pulses [32,57]. The safest method to avoid a precipitate from forming after use of NaOCl and CHX is to dispense only one of the two substances, which is the preferred option. Because of the sum of its properties, especially due to its ability to dissolve tissue, NaOCl is still the irrigation solution of choice during the mechanical preparation of the root canal. As long as the properties of potential alternatives, such as chlorine dioxide, have not been researched in more detail, it can still be regarded as the “gold standard” to use NaOCl exclusively in this phase [10].

In earlier studies, the combination of CHX and NaOCl was determined to have a better effect against Enterococcus faecalis and gram-positive germs compared to NaOCl alone [27,93,94,95]. Therefore, it was seen as an ideal complement to NaOCl. Some studies have since denied that the effectiveness of this combination against Enterococcus faecalis is better than that of NaOCl alone. However, this point is actually discussed controversially in the international literature [27,93,94,95,96].

A strict avoidance of the possible interaction between NaOCl and CHX in all its variants (including CHX solutions, CHX gels) is desired. According to the literature evaluated, this works best when an intermediate rinse with citric acid is utilized, as this removes the smear layer of the mechanical treatment without causing a precipitate with NaOCl or CHX and thereby triggering other complications.

In rinsing protocols that use NaOCl as the sole antimicrobial rinse, based on current knowledge, the final rinse to remove the smear layer should be carried out with citric acid or EDTA before the final use of NaOCl in an activated manner [10].

## 5. Conclusions

Since 2006, there has been a sharp increase in publications addressing the interactions between NaOCl and CHX. A total of 88 publications from the PubMed database were identified and evaluated. Of those, 58 publications were relevant to the topic. The results of the studies examined are often controversial, but certain aspects show a tendency over time.

The following findings relate to the endodontic irrigation protocol:-The chemo-mechanical preparation of the root canal system is currently the gold standard;-NaOCl should be used as the sole agent during mechanical reprocessing, due to its tissue-dissolving and antimicrobial properties;-The smear layer can be removed with CA or EDTA after the mechanical preparation. NaOCl should not be mixed with CA or EDTA, since chelators neutralize the tissue-dissolving effect of NaOCl;-The consecutive use of NaOCl and CHX is obsolete due to the precipitate that forms;-If NaOCl and CHX (or CHX derivatives) are used in the same tooth, intermediate rinsing is required. Since CHX also forms a precipitate with EDTA, CA is recommended for this.

These recommendations are useful in clinical practice to effectively avoid the formation of the undesirable precipitate.

## Figures and Tables

**Figure 1 antibiotics-12-00589-f001:**
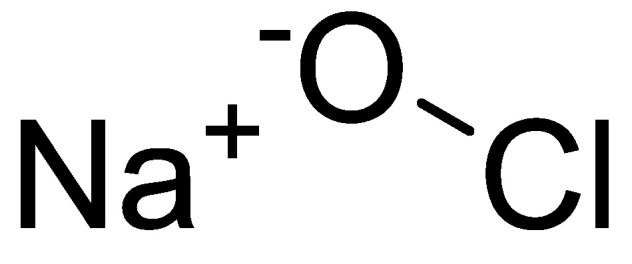
Structural formula of sodium hypochlorite.

**Figure 2 antibiotics-12-00589-f002:**
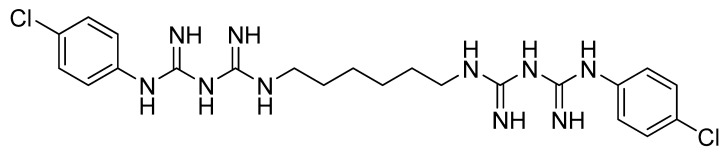
Structural formula of chlorhexidine gluconate.

**Figure 3 antibiotics-12-00589-f003:**
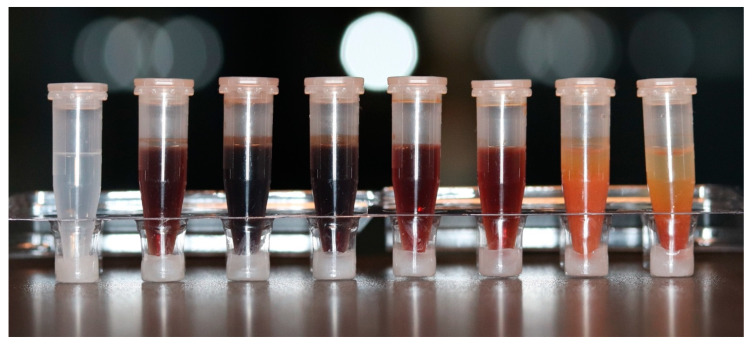
Microtubes containing 2% CHX mixed with different concentrations of NaOCl, to illustrate the precipitate formation. From left to right: (1) control sample with 0% NaOCl; (2) 0.5%; (3) 1%; (4) 1.5%; (5) 2.5%; (6) 3%; (7) 4%; (8) 5%.

**Figure 4 antibiotics-12-00589-f004:**
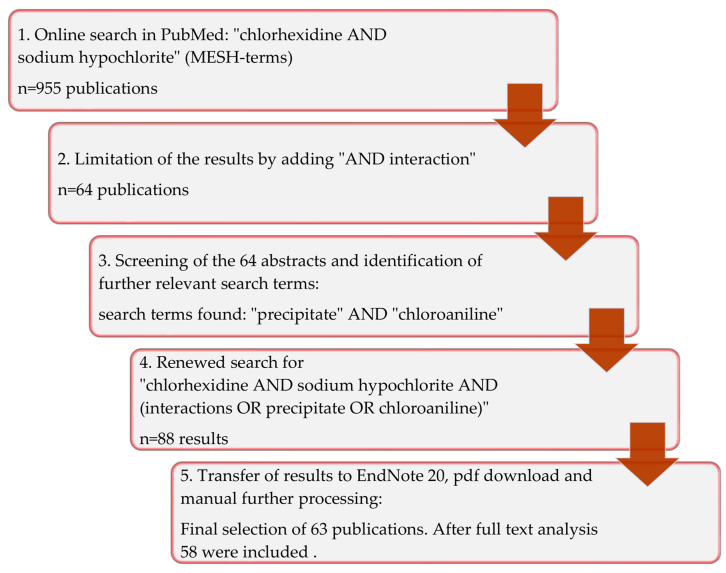
Graphical representation of the search strategy used in the present review.

**Table 1 antibiotics-12-00589-t001:** Inclusion and exclusion criteria.

Inclusion Criteria	Exclusion Criteria
- representing all search terms used	- articles without apparent relevance
- original articles, reviews, scientific short communications	- case reports, case series, editorials, case reviews

**Table 2 antibiotics-12-00589-t002:** Included papers of the review.

Author	Title	Study Aim	Type
Bueso et al., 2022 [34]	Comparative evaluation of intermediate solutions in prevention of brown precipitate formed from sodium hypochlorite and chlorhexidine gluconate	To evaluate intermediate treatments between sodium hypochlorite and chlorhexidine gluconate irrigations for the prevention of a toxic brown precipitate in root canal therapy.	Laboratory study
Jeong et al., 2021 [35]	Assessment of the cytotoxic effects and chemical composition of the insoluble precipitate formed from sodium hypochlorite and chlorhexidine gluconate	To investigate (1) the cytotoxic potential of the brown precipitate (BP) formed with sodium hypochlorite (NaOCl) and chlorhexidine gluconate (CHX), using both a small animal model of Caenorhabditis elegans (C. elegans) and cultured human gingival fibroblasts; (2) the chemical composition of BP using Time-of-Flight Secondary Ion Mass Spectrometry (ToF-SIMS).	Laboratory study
Czopik et al., 2021 [36]	Insight into the Reaction of Alexidine with Sodium Hypochlorite: A Potential Error in Endodontic Treatment	The aim of this study was to identify detected chemical compounds formed in the reaction of ALX and NaOCl with the ultra-high-performance liquid chromatography–mass spectrophotometry (UHPLC-MS) method and assess whether precipitates and PCA are formed in this reaction.	Laboratory study
Alberto et al., 2021 [37]	Does sodium thiosulphate avoid the formation of the brown-coloured precipitate as an intermediate irrigant between NaOCl and chlorhexidine?	This study evaluated the efficacy of sodium thiosulphate (ST) as an intermediate irrigant between sodium hypochlorite (NaOCl) and chlorhexidine (CHX) to avoid the formation of the brown-coloured precipitate.	Laboratory study
Khatib et al., 2020 [33]	Decoding the Perplexing Mystery of Para-Chloroaniline Formation: A Systematic Review	The purpose of this systematic review is to evaluate the relationship between PCA and brown precipitate.	Systematic review
Keles et al., 2020 [32]	Effect of various solutions on the removal of orange-brown precipitate formed by interaction of sodium hypochlorite and chlorhexidine with or without ultrasonic activation	The aim of this in vitro study was to investigate the possible interactions between photon-induced photoacoustic streaming (PIPS™)-activated oxidizing agents and 2% chlorhexidine digluconate.	Laboratory study
Buyukozer Ozkan et al., 2020 [38]	Proton Nuclear Magnetic Resonance Spectroscopy Analysis of Mixtures of Chlorhexidine with Different Oxidizing Agents Activated by Photon-Induced Photoacoustic Streaming for Root Canal Irrigation	The aim of the study was to assess the depth of sealer penetration into dentinal tubules following different final rinses and indirectly evaluate precipitation of irrigating solutions.	Laboratory study
Abusteit, 2020 [39]	Evaluation of resin sealer penetration of dentin following different final rinses for endodontic irrigation using confocal laser scanning microscopy	This study aimed to evaluate the characterization of chemical interaction of root canal irrigants on the surface of EndoSequence root repair materials using spectroscopy analysis.	Laboratory study
Abu Zeid et al., 2020 [40]	Morphological and chemical analysis of surface interaction of irrigant-endosequence root repair material	This study aimed to evaluate the characterization of chemical interaction of root canal irrigants on the surface of EndoSequence root repair materials using spectroscopy analysis.	Laboratory study
Thomas et al., 2019 [41]	Evaluation of the Antibacterial Efficiency of a Combination of 1% Alexidine and Sodium Hypochlorite on Enterococcus faecalis Biofilm Models: An In Vitro Study	The aim of the study was to assess the antibacterial efficiency of a combination of 1% alexidine (ALX) and 5.25% sodium hypochlorite (NaOCl) against E. faecalis biofilm using a confocal scanning electron microscopy.	Laboratory study
Siddique et al., 2019b [16]	Qualitative and quantitative analysis of precipitate formation following interaction of chlorhexidine with sodium hypochlorite, neem, and tulsi	This study aims to evaluate the precipitate formed on combination of different irrigants, weigh the amount of precipitate formed and to analyze the precipitate for PCA.	Laboratory study
Siddique et al., 2019a [42]	Quantitative analysis for detection of toxic elements in various irrigants, their combination (precipitate), and para-chloroaniline: An inductively coupled plasma mass spectrometry study	The aim of this study was to evaluate the precipitate formed on combination of different irrigants, weigh the amount of precipitate formed, and to analyze 35 different metal elements in each irrigant, precipitate formed as well as in PCA.	Laboratory study
Žižka et al., 2018 [43]	Discoloration after Regenerative Endodontic Procedures: A Critical Review	This review presents a critical view on current knowledge of discoloration sources, its treatment and possible preventive modalities.	review
Ravinanthanan et al., 2018 [44]	Cytotoxicity Evaluation of Combination Irrigant Regimens with MTAD on Two Different Cell Lines	The aim of this study was to evaluate the cytotoxicity of combination regimens on target and nontarget cell lines by trypan blue assay.	Laboratory study
Piperidou et al., 2018 [45]	Effects of Final Irrigation with SmearOFF on the Surface of Dentin Using Surface Analytical Methods	This study examined the chemical interaction of SmearOFF with sodium hypochlorite (NaOCl) on the dentin surface, specifically the formation of precipitate and/or parachloroaniline (PCA).	Laboratory study
Jain et al., 2018 [46]	Alexidine versus chlorhexidine for endodontic irrigation with sodium hypochlorite	The objective of this study was to chemically evaluate precipitate formation on irrigation by different concentrations of chlorhexidine (CHX) and alexidine (ALX) with sodium hypochlorite (NaOCl).	Laboratory study
Irmak et al., 2018 [47]	Nuclear magnetic resonance spectroscopy and infrared spectroscopy analysis of precipitate formed after mixing sodium hypochlorite and QMix 2 in 1	This study assessed whether para-chloroaniline (PCA) is formed after mixing NaOCl with Qmix.	Laboratory study
Gonzalez et al., 2018 [48]	Temperature changes in 2% chlorhexidine gluconate using two activation methods with different intensity levels	…the objective is to establish the influence of ultrasonic and sonic activation, with the use of different intensities, upon the temperature of chlorhexidine gluconate (CHX).	Laboratory study
Chhabra et al., 2018 [15]	Efficacy of various solutions in preventing orange-brown precipitate formed during alternate use of sodium hypochlorite and chlorhexidine: An In vitro study	The study evaluated the effectiveness of three Intermediate endodontic irrigating solutions in eliminating the residual sodium hypochlorite (NaOCl).	Laboratory study
Campbell et al., 2018 [49]	Antiseptics Commonly Used in Total Joint Arthroplasty Interact and May Form Toxic Products	Our clinical experience is that chlorhexidine (CHX) and Dakin’s solution (NaOCl) interact and form a precipitate. The purpose of this study is to determine whether this reaction could be replicated in a laboratory setting, and to determine if other commonly used antiseptics also visibly react when mixed.	Laboratory study
Wright et al., 2017 [7]	Alkaline Sodium Hypochlorite Irrigant and Its Chemical Interactions	Of particular interest is the interaction between sodium hypochlorite and the chelators EDTA, citric acid and etidronate and between sodium hypochlorite and the antimicrobials chlorhexidine, alexidine, MTAD and octenisept.	review
Thaha et al., 2017 [50]	Interaction between Octenidine-based Solution and Sodium Hypochlorite: A Mass Spectroscopy, Proton Nuclear Magnetic Resonance, and Scanning Electron Microscopy-based Observational Study	The aim of this study was first to Identify the precipitate formed on the interaction between OCT and NaOCl and secondly to compare its effect on dentinal tubules with that of precipitate formed on combining chlorhexidine (CHX) and NaOCl.	Laboratory study
Surrender et al., 2017 [51]	Alexidine: A Safer and an Effective Root Canal Irrigant than Chlorhexidine	AIM: To compare antimicrobial activity of different concentrations of ALX with CHX individually and when combined with NaOCl against E. faecalis strains.	Laboratory study
Nocca et al., 2017 [14]	Chromographic Analysis and Cytotoxic Effects of Chlorhexidine and Sodium Hypochlorite Reaction Mixtures	This study aimed to investigate the stability of PCA in the presence of NaOCl and to examine the in vitro cytotoxic effects of CHX/NaOCl reaction mixtures.	Laboratory study
Guneser et al., 2017 [52]	Comparison of Conventional Syringe, CanalBrush, EndoActivator, Photon-Induced Photoacoustic Streaming, and Manual Instrumentation in Removing Orange-Brown Precipitate: An In Vitro Study	The aim of this In vitro study was to compare the various techniques for removing precipitate formed after irrigation with sodium hypochlorite (NaOCl) and chlorhexidine (CHX).	Laboratory study
Vouzara et al., 2016 [53]	Combined and independent cytotoxicity of sodium hypochlorite, ethylenediaminetetraacetic acid and chlorhexidine	AIM: To evaluate the capacity of commonly used root canal irrigants to induce cytotoxic effects, when applied singly or in combination	Laboratory study
Patil et al., 2016 [54]	Determination of mutagenicity of the precipitate formed by sodium hypochlorite and chlorhexidine using the Ames test	The aim of this study was to determine the direct mutagenic potential of any precipitate formed by combining sodium hypochlorite (NaOCl) and chlorhexidine (CHX).	Laboratory study
Orhan et al., 2016 [55]	Does Para-chloroaniline Really Form after Mixing Sodium Hypochlorite and Chlorhexidine?	Purpose of this study was to determine whether PCA is formed through the reaction of mixing NaOCl and CHX.	Laboratory study
Mohammadi et al., 2015 [56]	Agonistic and Antagonistic Interactions between Chlorhexidine and Other Endodontic Agents: A Critical Review	The aim of this investigation was to review the agonistic and antagonistic interactions between chlorhexidine (CHX) and other irrigants and medicaments.	review
Metri et al., 2015 [57]	Comparative Evaluation of Two Final Irrigation Techniques for the Removal of Precipitate Formed by the Interaction between Sodium Hypochlorite and Chlorhexidine	AIM: To evaluate the effectiveness of two final irrigation techniques for the removal of precipitate formed by the interaction between sodium hypochlorite (NaOCl) and chlorhexidine (CHX).	Laboratory study
Magro et al., 2015 [58]	Effectiveness of several solutions to prevent the formation of precipitate due to the interaction between sodium hypochlorite and chlorhexidine and its effect on bond strength of an epoxy-based sealer	AIM: To evaluate the effectiveness of isopropyl alcohol, saline or distilled water to prevent the precipitate formed between sodium hypochlorite (NaOCl) and chlorhexidine (CHX) and its effect on the bond strength of an epoxy-based sealer in radicular dentine.	Laboratory study
Bernardi & Teixeira, 2015 [28]	The properties of chlorhexidine and undesired effects of its use in endodontics	The purpose of this article was to review the literature on the properties of chlorhexidine (CHX) and the adverse effects that may occur from its use in endodontics.	review
Arslan et al., 2015 [59]	Evaluation of orange-brown precipitate formed in root canals after irrigation with chlorhexidine and QMix and spectroscopic analysis of precipitates produced by a mixture of chlorhexidine/NaOCl and Qmix/NaOCl	AIM: To compare chlorhexidine and Qmix™ in terms of orange-brown precipitate generation in root canals and (ii) to analyse the precipitate produced by mixing chlorhexidine and Qmix(™) with NaOCl to determine whether para-chloroaniline was produced.	Laboratory study
Kolosowski et al., 2014 [60]	Qualitative analysis of precipitate formation on the surface and in the tubules of dentin irrigated with sodium hypochlorite and a final rinse of chlorhexidine or QMiX	The aim of this study was to qualitatively assess the formation of precipitate and PCA on the surface and in the tubules of dentin irrigated with NaOCl, followed either by EDTA, NaOCl, and CHX or by saline and QMiX.	Laboratory study
Homayouni et al., 2014 [61]	The Effect of Root Canal Irrigation with Combination of Sodium Hypo-chlorite and Chlorhexidine Gluconate on the Sealing Ability of Obturation Materials	The aim of this study was to evaluate the effect of the precipitate that was formed by combining Sodium Hypochlorite (NaOCl) and Chlorhexidine Gluconate (CHX) on the sealing ability of root canal obturation materials.	Laboratory study
Magro et al., 2014 [62]	Evaluation of the interaction between sodium hypochlorite and several formulations containing chlorhexidine and its effect on the radicular dentin—SEM and push-out bond strength analysis	The aim of the current study was to evaluate the presence of debris and smear layer after endodontic irrigation with different formulations of 2% chlorhexidine gluconate (CHX) and its effects on the push-out bond strength of an epoxy-based sealer on the radicular dentin.	Laboratory study
Cintra et al., 2014 [63]	The use of NaOCl in combination with CHX produces cytotoxic product	The aim of this study was to evaluate the tissue response to implanted polyethylene tubes filled with PPT-soaked fibrin sponge.	Laboratory study
Arslan et al., 2014 [64]	Evaluation of effectiveness of various irrigating solutions on removal of calcium hydroxide mixed with 2% chlorhexidine gel and detection of orange-brown precipitate after removal	The aims of the present study were to evaluate the effect of various irrigating solutions on the removal of calcium hydroxide mixed with 2% chlorhexidine gel from an artificial groove created in a root canal and the generation of orange-brown precipitate in the remaining calcium hydroxide mixed with 2% chlorhexidine gel after irrigation with the various irrigating solutions.	Laboratory study
Souza et al., 2013 [65]	Evaluation of the colour change in enamel and dentine promoted by the interaction between 2% chlorhexidine and auxiliary chemical solutions	AIM: To evaluate the colour change in enamel and dentine, promoted by interaction of 2% chlorhexidine gluconate (CHX) with 5.25% sodium hypochlorite (NaOCl) and 17% ethylenediaminetetraacetic acid (EDTA).	Laboratory study
Shenoy et al., 2013 [66]	Assessment of precipitate formation on interaction of irrigants used in different combinations: an in vitro study	AIM: To evaluate the combination of various irrigants whether it forms the precipitate and also to quantify the amount of precipitate formed.	Laboratory study
Rossi-Fedele et al., 2013 [67]	Interaction between chlorhexidine-impregnated gutta-percha points and several chlorine-containing endodontic irrigating solutions	AIM: To evaluate if the immersion of chlorhexidine-impregnated gutta-percha points in chlorine-containing endodontic irrigants causes colour changes and precipitate formation.	Laboratory study
Prado et al., 2013 [68]	Interactions between irrigants commonly used in endodontic practice: a chemical analysis	The aim of this work was to characterize the by-products formed in the associations between the most commonly used irrigants in endodontic practice.	Laboratory study
Pasich et al., 2013 [69]	Efficacy of taurine haloamines and chlorhexidine against selected oral microbiome species	In this in vitro study we have compared antimicrobial activity of CHX with that of taurine chloramine (TauC1) and taurine bromamine (TauBr).	Laboratory study
Gupta et al., 2013 [70]	Evaluation of the sealing ability of two sealers after using chlorhexidine as a final irrigant: An in vitro study	The aim of this study was to evaluate the effect of the precipitate formed by using sodium hypochlorite and chlorhexidine as a root canal irrigant on the sealing ability of different root canal sealers.	Laboratory study
Gomes et al., 2013 [6]	Chlorhexidine in endodontics	The aim of this paper is to review CHX’s general use in the medical field and in dentistry.	review
Vilanova et al., 2012 [71]	Effect of intracanal irrigants on the bond strength of epoxy resin-based and methacrylate resin-based sealers to root canal walls	AIM: To assess the bond strength of Epiphany and AH Plus sealers to root canal walls using a push-out test after use of several endodontic irrigants.	Laboratory study
Rossi-Fedele et al., 2012 [72]	Antagonistic interactions between sodium hypochlorite, chlorhexidine, EDTA, and citric acid	The aim of this investigation was to review the antagonistic interactions occurring when sodium hypochlorite (NaOCl), chlorhexidine (CHX), EDTA, and citric acid (CA) are used together during endodontic treatment.	review
Mortenson et al., 2012 [13]	The effect of using an alternative irrigant between sodium hypochlorite and chlorhexidine to prevent the formation of para-chloroaniline within the root canal system	AIM: To determine if the formation of para-chloroaniline (PCA) can be avoided by using an alternative irrigant following sodium hypochlorite but before chlorhexidine.	Laboratory study
Kim, 2012 [31]	Precipitate from a combination of sodium hypochlorite and chlorhexidine	…Chlorhexidine can form a precipitate when used in combination with NaOCl during intra-canal irrigation. What is the adverse effect of this precipitate and how can I reduce the chance of precipitation?	short communication
Kim et al., 2012 [73]	Chemical interaction of alexidine and sodium hypochlorite	This study determined by electrospray ionization mass spectrometry (ESI-MS) and scanning electron microscopy (SEM) whether the chemical interaction between ALX and NaOCl results in PCA or precipitates.	Laboratory study
Gasic et al., 2012 [74]	Ultrastructural analysis of the root canal walls after simultaneous irrigation of different sodium hypochlorite concentration and 0.2% chlorhexidine gluconate	AIM: To determine whether sodium hypochlorite (NaOCl) with 0.2% chlorhexidine gluconate (CHX) leads to colour change and precipitate formation, and to ultrastructurally analyse the dentine surface after simultaneous irrigation with 0.5% NaOCl and 0.2% CHX.	Laboratory study
Prado et al., 2011 [75]	Effect of disinfectant solutions on the surface free energy and wettability of filling material	The aims of this study were to evaluate the surface free energy of GP and Res cones after disinfection procedures and to investigate the wettability of endodontic sealers in contact with these surfaces.	Laboratory study
Nowicki & Sem, 2011 [76]	An in vitro spectroscopic analysis to determine the chemical composition of the precipitate formed by mixing sodium hypochlorite and chlorhexidine	The purpose of this in vitro study was to determine the chemical composition of the precipitate formed by mixing sodium hypochlorite (NaOCl) and chlorhexidine (CHX) and the relative molecular weight of the components.	Laboratory study
de Assis et al., 2011 [77]	Evaluation of the interaction between endodontic sealers and dentin treated with different irrigant solutions	The aim of this study was to investigate the wettability of endodontic sealers in contact with dentin treated with 5.25% sodium hypochlorite (NaOCl) and 2% chlorhexidine (CHX) in the presence or absence of smear layer.	Laboratory study
Thomas & Sem, 2010 [78]	An in vitro spectroscopic analysis to determine whether para-chloroaniline is produced from mixing sodium hypochlorite and chlorhexidine	The purpose of this in vitro study was to determine whether para-chloroaniline (PCA) is formed through the reaction of mixing sodium hypochlorite (NaOCl) and chlorhexidine (CHX).	Laboratory study
Krishnamurthy & Sudhakaran, 2010 [79]	Evaluation and prevention of the precipitate formed on interaction between sodium hypochlorite and chlorhexidine	The purpose of this study was (1) to evaluate maximum thickness the and chemical composition of the precipitate formed between sodium hypochlorite (NaOCl) and chlorhexidine (CHX) and (2) to evaluate effectiveness of absolute alcohol to remove residual NaOCl and thereby prevent the formation of the precipitate.	Laboratory study
Basrani et al., 2010 [80]	Determination of 4-chloroaniline and its derivatives formed in the interaction of sodium hypochlorite and chlorhexidine by using gas chromatography	The aim of this study was to further identify the precipitate by using gas chromatography-mass spectrometry (GC-MS).	Laboratory study
Akisue et al., 2010 [12]	Effect of the combination of sodium hypochlorite and chlorhexidine on dentinal permeability and scanning electron microscopy precipitate observation	This study compared the combined use of sodium hypochlorite (NaOCl) and chlorhexidine (CXH) with citric acid and CXH on dentinal permeability and precipitate formation.	Laboratory study
Mohammadi & Abbott, 2009 [81]	The properties and applications of chlorhexidine in endodontics	The purpose of this paper is to review the structure and mechanism of action of CHX, its antibacterial and antifungal activity, its effect on biofilm, its substantivity (residual antibacterial activity), its tissue solvent ability, its interaction with calcium hydroxide and sodium hypochlorite, its anticollagenolytic activity, its effect on coronal and apical leakage of bacteria, its toxicity and allergenicity and the modulating effect of dentine and root canal components on its antimicrobial activity.	review
Basrani et al., 2009 [82]	Using diazotization to characterize the effect of heat or sodium hypochlorite on 2.0% chlorhexidine	The aim of the present study was to use a diazotization technique to confirm the presence of an aromatic amine (such as PCA) in the NaOCl/CHX precipitate and also in the 2.0% CHX at different temperatures (37 degrees C and 45 degrees C).	Laboratory study
Bui et al., 2008 [30]	Evaluation of the interaction between sodium hypochlorite and chlorhexidine gluconate and its effect on root dentin	The aim of this study was to evaluate the effect of irrigating root canals with a combination of NaOCl and CHX on root dentin and dentinal tubules…	Laboratory study
Basrani et al., 2007 [11]	Interaction between sodium hypochlorite and chlorhexidine gluconate	The aim of this study was to determine the minimum concentration of NaOCl required to form a precipitate with 2.0% CHX.	Laboratory study
Zehnder, 2006 [5]	Root canal irrigants	In this review article, the specifics of the pulpal microenvironment and the resulting requirements for irrigating solutions are spelled out.	review

## Data Availability

Not applicable.

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
