# Peer review of "The Interaction of Two Widely Used Endodontic Irrigants, Chlorhexidine and Sodium Hypochlorite, and Its Impact on the Disinfection Protocol during Root Canal Treatment"

_antibiotics, 2023, doi:10.3390/antibiotics12030589_

Round 1
Reviewer 1 Report
The topic is interesting and fits well the scope of Antibiotics. The reviewer has no objection to publish this review article.
(1) The authors may briefly discuss the mechanism of cytotoxicity of the precipitate
(2) The authors may briefly how to amend the clinical practice guideline to avoid such interaction.
Author Response
Thank you very much for your time, work and comments. They are very helpful to improve the manuscript.
The point-by- point response:
(1) The authors may briefly discuss the mechanism of cytotoxicity of the precipitate
This point was addressed and added in the discussion section.
(2) The authors may briefly how to amend the clinical practice guideline to avoid such interaction.
Thank you for this comment. This point was also adressed and included in the discussion and conclusion section. A really important point.
Reviewer 2 Report
The authors should add a comparison table of protocol comparisons, to analyze the significant variations between protocol, and their possible outcomes.
In Table 1, the authors may include the column of "Main study outcome" and "Study Aim", while the detailed author name, and title name can be removed, since those information can be later found in references list.
In the conclusion, "The following results relate to the endodontic irrigation protocol" The results list can be turned into a table in results section.
As of Figure 3, please include the O.D. readings of colorimetric comparisons, whenever available.
The authors may include all relevant literatures of the captioned topic.
The authors can give details of the inclusion and exclusion criteria in the methodology.
The authors can make a list of consideration and/or caution, of this captioned topic (in a figure).
In Table 1, the authors can add a column to categorise whether the included publications, regarded as, preclinical animal, or in vitro, clinical study (stages and N number). Please sort the list in Table 1, by years of publications, or the above mentioned groupings.
The authors should subdivide the discussion section, into several subsection, i.e. 4.1, 4.2, 4.3, i.e. by toxicity.
Author Response
First of all, thank you very much for the review and all the helpful comment to improve the manuscript.
The point-by- point response:
The authors should add a comparison table of protocol comparisons, to analyze the significant variations between protocol, and their possible outcomes.
Due to the remarkable number of different protocols and studies, we decided to address the differences in the discussion section. An additional table with all protocols would exceed the length of the review and decrease the readability of the paper.
In Table 1, the authors may include the column of "Main study outcome" and "Study Aim", while the detailed author name, and title name can be removed, since those information can be later found in references list.
Thank you very much for your comment. The main outcomes are included in the discussion section. Additionally, we include more information about the included studies in the table. We have based the formatting on previous reviews published in Antibiotics and designed our table accordingly.
In the conclusion, "The following results relate to the endodontic irrigation protocol" The results list can be turned into a table in results section.
We have formulated our recommendations for preventing the formation of a precipitate in the summary. Thank you for this comment, this point is very important.
As of Figure 3, please include the O.D. readings of colorimetric comparisons, whenever available.
The image is a visualization to provide a visual representation of the topic for better understanding to readers. No colorimetric measurements were performed. It was made just for illustration purpose.
The authors may include all relevant literatures of the captioned topic.
Thank you for your advice. We have included all papers related to the named search terms from 1974-2022.
The authors can give details of the inclusion and exclusion criteria in the methodology.
Thank you very much for your advice. We have added a table with the inclusion and exclusion criteria.
The authors can make a list of consideration and/or caution, of this captioned topic (in a figure).
We have formulated our recommendations for preventing the formation of a precipitate in the discussion section and conclusion. Thank you for this comment, this point is very important.
In Table 1, the authors can add a column to categorise whether the included publications, regarded as, preclinical animal, or in vitro, clinical study (stages and N number). Please sort the list in Table 1, by years of publications, or the above mentioned groupings.
Thank you very much for your comment. We have included more information about the cited studies. No clinical studies were found related to the topic. Based the formatting on previous reviews in the journal and designed our table accordingly. The order of our table is sorted according to the years of publication.
The authors should subdivide the discussion section, into several subsection, i.e. 4.1, 4.2, 4.3, i.e. by toxicity.
Thank you very much for your comment. We have subdivided the discussion section according to your preferences.
Reviewer 3 Report
overall the paper is written good in all aspects. Very minor mistakes such as re-check the spelling and grammer. Further, Some of references are not according to the jounral style. Re-check again. Further, some mentioend of standards in paper, like according to which guidelines , these two chlorhexidine and sodium hypochlorite are recommended. Further, how we can reduce the chances of these interactrions and what will be the other options.
Author Response
First of all, thank you very much for the helpful comments. We could adress all comments.
The point-by- point response:
Overall the paper is written good in all aspects. Very minor mistakes such as re-check the spelling and grammer.
The spelling and grammer was checked and corrected if needed.
Further, Some of references are not according to the jounral style. Re-check again.
All references were checked and corrected according to the journal style.
Further, some mentioend of standards in paper, like according to which guidelines , these two chlorhexidine and sodium hypochlorite are recommended.
The guidelines were added and included in the references. Thank you very much for this comment. This is really an important point.
Further, how we can reduce the chances of these interactrions and what will be the other options.
This point was added in the discussion and conclusion section. Thank you very much for the support and the help in improving the paper.